# Community interventions in Low—And Middle-Income Countries to inform COVID-19 control implementation decisions in Kenya: A rapid systematic review

Leila Abdullahi[1]*, John Joseph Onyango[2], Carol Mukiira[1], Joyce Wamicwe[2], Rachel Githiomi[2], David Kariuki[2], Cosmas Mugambi[2], Peter Wanjohi[2], George Githuka[2], Charles Nzioka[2], Jennifer Orwa[3], Rose Oronje[1], James Kariuki[3], Lilian Mayieka[3]

1 African Institute for Development Policy (AFIDEP), Nairobi, Kenya, 2 Directorate of Research, Ministry of Health (MOH), Nairobi, Kenya, 3 Knowledge Management Unit, Kenya Medical Research Institute (KEMRI), Nairobi, Kenya

* Leyla.abdullahi@afidep.org

## Abstract

Globally, public health measures like face masks, hand hygiene and maintaining social distancing have been implemented to delay and reduce local transmission of COVID-19. To date there is emerging evidence to provide effectiveness and compliance to intervention measures on COVID-19 due to rapid spread of the disease. We synthesized evidence of community interventions and innovative practices to mitigate COVID-19 as well as previous respiratory outbreak infections which may share some aspects of transmission dynamics with COVID-19. In the study, we systematically searched the literature on community interventions to mitigate COVID-19, SARS (severe acute respiratory syndrome), H1N1 Influenza and MERS (middle east respiratory syndrome) epidemics in PubMed, Google Scholar, World Health Organization (WHO), MEDRXIV and Google from their inception until May 30, 2020 for up-to-date published and grey resources. We screened records, extracted data, and assessed risk of bias in duplicates. We rated the certainty of evidence according to Cochrane methods and the GRADE approach. This study is registered with PROSPERO (CRD42020183064). Of 41,138 papers found, 17 studies met the inclusion criteria in various settings in Low- and Middle-Income Countries (LMICs). One of the papers from LMICs originated from Africa (Madagascar) with the rest from Asia 9 (China 5, Bangladesh 2, Thailand 2); South America 5 (Mexico 3, Peru 2) and Europe 2 (Serbia and Romania). Following five studies on the use of face masks, the risk of contracting SARS and Influenza was reduced OR 0.78 and 95% CI = 0.36–1.67. Equally, six studies on hand hygiene practices reported a reduced risk of contracting SARS and Influenza OR 0.95 and 95% CI = 0.83–1.08. Further two studies that looked at combined use of face masks and hand hygiene interventions showed the effectiveness in controlling the transmission of influenza OR 0.94 and 95% CI = 0.58–1.54. Nine studies on social distancing intervention demonstrated the importance of physical distance through closure of learning institutions on the transmission dynamics of disease. The evidence confirms the use of face masks, good hand hygiene and social distancing as community interventions are effective to control the spread of SARS and

**Data Availability Statement:** All relevant data are within the paper and its Supporting information files.

**Funding:** The study is funded under Heightening Institutional Capacity for Government Use of Health Research (HIGH-Res) Project. The HIGH-Res project received financial support from the Alliance for Health Policy and Systems Research (the Alliance) at the World Health Organisation, and Wellcome Trust. The funders had no role in study design, data collection and analysis, decision to publish, or preparation of the manuscript.

**Competing interests:** The authors have declared that no competing interests exist.

**Abbreviations:** CBAs, Controlled before-and-after studies; CDC, Centers for Disease Control; COVID-19, Coronavirus Disease 2019; cRCT, Cluster-Randomised Controlled Trials; GAVI, Global Alliance for Vaccines and Immunization; HIGH-Res, Heightening Institutional Capacity for Government Use of Health Research; HIN1, Influenza A virus subtype H1N1 (A/H1N1); ILI, Influenza-Like Illness; ITS, Interrupted time series; LMIC, Low- and Middle- Income Countries; MERS-CoV, The Middle East Respiratory Syndrome Coronavirus; MoH, Ministry of Health; RCT, Randomised Controlled Trials; SARs, Severe Acute Respiratory Syndrome; SARS-CoV, Severe Acute Respiratory Syndrome Coronavirus; UNICEF, United Nations Children's Fund; WHO, World Health Organisation.

influenza in LMICs. However, the effectiveness of community interventions in LMICs should be informed by adherence of the mitigation measures and contextual factors taking into account the best practices. The study has shown gaps in adherence/compliance of the interventions, hence a need for robust intervention studies to better inform the evidence on compliance of the interventions. Nevertheless, this rapid review of currently best available evidence might inform interim guidance on similar respiratory infectious diseases like Covid-19 in Kenya and similar LMIC context.

## List of definitions

1. Contact tracing: Identification and follow-up of persons who may have had contact with a person infected with COVID-19. The contact tracing process involves four main steps:

   - Contact identification
   - Contact listing and classification
   - Contact monitoring
   - Contact discharge

2. Social distancing: Term applied to public health measures taken to delay and diminish transmission of COVID-19.

   - At the individual level, social distancing involves the use of non-contact greetings, maintaining at least one metre distance between yourself and other people, and staying home when ill.

   - At the community level, social distancing involves closure of any events or settings in which people gather together, including schools, workplaces, houses of worship, and cultural, social and sports events.

3. Isolation: Refers to the separation of people with symptoms (i.e. sick people) to prevent spread of the infection to healthy individuals.

4. Quarantine: For COVID-19 public health practice, quarantine refers to separating and restricting the movement of a healthy (i.e. non-infected) person who is at risk of COVID-19.

5. Influenza-Like Illness (ILI): Acute respiratory infection with measured fever of $\geq$ 38 C˚,

6. and cough with onset within the last 10 days.

7. Severe Acute Respiratory syndrome (SARs): An acute respiratory presentation with a history of fever or measured fever of $\geq$ 38 C˚, and cough, with onset within the last 10 days, and requires hospitalization.

8. An N95 respirator is a respiratory protective device designed to achieve a very close facial fit.

Definitions were adapted from Africa Centers for Disease Control (CDC) study on recommendations-Stepwise-Response-COVID-19 [1].

## Background

### Description of the condition

The World Health Organization (WHO) declared the Coronavirus Disease 2019 (COVID-19) a pandemic on March 11, 2020 [2]. The rapid spread of the COVID-19 outbreak with over 2 million cases has had great global impact [3] whose ripple effect to Kenya is loudly felt. Coronavirus belongs to the family of viruses that cause viral pneumonia and include symptoms such as fever, breathing difficulty, and lung infection [4]. Globally, countries are using various measures to curb the pandemic. These measures include complete and/or partial lockdowns, shifting to remote working, online schooling, promoting regular hand-washing, use of masks, and social distancing. Most of these recommended measures have been implemented in high-income countries but may not necessarily work for Low- and Middle-Income country (LMIC) contexts [5].

Coronavirus is a new pathogen with no pharmaceutical intervention, therefore to slow down the spread of the virus, community mitigations otherwise categorized as non-pharmaceutical interventions are the options available [6]. Recommended COVID-19 community measures include hand hygiene, coughing etiquette, use of masks, and social distancing. In LMICs, implementation of the recommended community measures is a challenge due to conditions of vulnerability peculiar to these populations that need to be taken into consideration. For example, street dwellers, people living in overcrowded households/slums, households without adequate ventilation or without running water, migrants and refugee settings [7, 8]. In the past years, LMICs have had experience dealing with different epidemics like Ebola, polio and cholera [9, 10]. Drawing lessons learnt on preparedness and response to previous epidemics with respect to community measures and control is thus crucial in enabling African countries utilize effective innovative strategies to curb the spread of COVID-19.

### Description of the intervention

**Social distancing.** Social distancing is the term used for non-pharmaceutical measures that reduce physical contact between infectious and susceptible people during a disease outbreak. Epidemiological and modeling studies have showed reduced viral spreading rate and evidence of delayed epidemic peaks with social distance intervention [11]. Evidence suggests that social and physical distancing are more effective when combined with other interventions. Broadly social distancing is categorized into two types:

- The first aims to prevent the transmission of virus from infectious individuals to others who come into close contact.

- The second set of measures aims to stop people from meeting at all, for example by closing schools, shops, and workplaces, banning mass gatherings, advising people to stay at home, and suspending public transport.

The challenge is to find out which forms of social distancing work best for various settings / geographical areas taking into account the limited space in LMIC setting.

**Face masking.** Face masking is the creation of a barrier around the breathing zones in order to break the chain of transmission by reducing the infectiousness of the virus shedder, thus offering protection to the susceptible individual [12]. Use of face masks as source control has been shown to decrease the release of respiratory droplets when coughing, talking and even breathing. Masks offer protection by inhibiting the expelled particle from being projected forward as a rapid jet over a distance to reach the breathing zone of the susceptible individual, instead the particle is decelerated or redirected.

Community-wide face masking has been shown to control the incidence of COVID-19 in Hong Kong Special Administration region compared to other countries where community wide face masking was hindered by other practices due to religion and behavior [13, 14]. During epidemics, there is a high number of asymptomatic cases, despite the viral shedding, these cases cannot be recognized unless they seek medical attention [13]. Face masking is therefore a crucial measure. It serves as means of source control though may create a sense of false protection which might lead to relaxation of other measures like social distancing and respiratory etiquette [12]. Personal responsibility should therefore be observed to ensure total protection. Wearing face masks is especially recommended when visiting public places, using public transport and for crowded work places [14].

There exist several types of masks; surgical masks, N95 and homemade masks. All have been demonstrated to offer protection with the N95 being the most superior offering 50 times as much protection as homemade masks and 25 times as much protection as surgical masks [15]. For maximum protection masks should be worn appropriately, ensuring the hands have been cleaned with soap and water or alcohol-based sanitizer [16]. Masks should cover the face from the bridge of the nose down to the chin. When handling masks, they should be safely removed from the back without touching the front, then safely disposed. Washable masks should be washed immediately with a the detergent [16, 17].

**Hand hygiene.**   Hand hygiene is one of the most effective preventive measures [18] because hands are the main pathway of COVID-19 germs transmission into the human respiratory system. In October 2005, WHO launched the first global safety challenge called 'Cleaner Care Safer Care' whose key action was to promote hand hygiene globally and at all levels of health-care. One of the program's key accomplishment has been the development of hand hygiene guidelines [19]. Hand washing in the community has been documented to be highly effective in prevention of both diarrheal and respiratory illnesses [18], making it one of the most important mitigation measures.

Washing hands with water and soap is the best way to get rid of germs [20], but if it is not available, use of a 60% alcohol-based hand sanitizer is recommended. Hand sanitizers reduce the number of germs in the hands, however, they do not get rid of all types of germs, they may not work on visibly dirty hands, and might not remove harmful chemicals like pesticides from the hands [21]. Use of plain soap is effective at inactivating enveloped viruses, such as COVID-virus, by dissolving the membrane hence killing the virus [20].

The Centers for Disease Control (CDC) recommends five steps of washing hands correctly. This involves wetting your hands with clean, running water, lathering by rubbing them together with the soap, scrubbing your hands for at least 20 seconds while paying attention to the folds, nails and back of the hand, and rinsing your hands well with clean water then dry them. When using hand sanitizers, apply the gel on the palm of your hand, rub hands together until it dries up, this should take about 20 seconds [21].

## How the intervention work

Various interventions have been implemented globally to combat COVID-19 and other contagious diseases like Bovine Spongiform Encephalitis in 1986, the Avian flu in 1997, the SARS in 2002, the Swine Flu in 2009, and Ebola in 2014. Some of the interventions in place include complete and/or partial lockdowns, ban on gatherings leading to a shift to remote working and online schooling, promoting hand-washing, use of masks, and social distancing. The main objective of the interventions is to prevent transmission within the community, thereby flattening the peak of the disease [22] to ease pressure on the health care system as the development of treatment and vaccines for the virus is in the pipeline.

Social distancing has been used in previous pandemics and the experience might guide the world on what form of social distancing might work for COVID-19. Most of social distancing evidence comes from influenza, including the 1918–20 'Spanish flu' and the less extensive, but more recent 2009 swine flu [23]. Unlike influenza and COVID-19, Ebola is not a respiratory disease, but the 2014–15 West African Ebola outbreak offers lessons on social distancing as well [24].

Face masking has previously been used during respiratory outbreaks like the 2009 influenza A outbreak, Severe Acute Respiratory Syndrome coronavirus (SARS-CoV) [25] and the Middle East Respiratory Syndrome coronavirus (MERS-CoV). The effectiveness of masks in prevention of human to human transmission of respiratory infections has been demonstrated in two systematic reviews [26, 27].

Hands are the main vehicles of infection transmission because of the surface contact effect. Hand hygiene is therefore paramount and considered the easiest and cheapest way of infection control mitigation [19]. Washing hands with soap and water breaks away the infectious viruses hence reducing transmission [20, 28].

## Why it is important to do this review

Like most countries globally, Kenya, through the Ministry of Health (MoH) has set out several prevention and mitigation policies and interventions. Some of the interventions and policies includes social distancing, hand hygiene policy, mandatory use of face masks in public places, dusk to dawn curfew, closure of schools and any social and religious gathering places, cessation of movement in and out of Covid-19 hot spot areas, international travel bun, isolation of infected and exposed individual and currently the implementation of mass testing of all citizens. In Kenya, healthy individual with the infection also known as asymptomatic cases are estimated to be at about 60% [29, 30]. Asymptomatic individuals play a big role in the infection transmission hence community measures play a big role to curb rapid spread of the epidemic.

As of July 2020, the Kenyan Ministry of Health in the daily Kenyan Government briefing reported a rise in the number of cases per day [31] despite the mitigation implementation. This poses a question of compliance, what are the factors that affect compliance and how can these be controlled. Currently, there is emerging evidence to understand barriers and enablers of the best practices, lessons and innovative community measures in place to curb the spread of COVID-19. For example, prohibiting mass gathering to observe social distancing measure through closure of learning institution and religious and social places has been implemented in Kenya with an aim of curbing the spread of COVID-19. Fong et al identified three studies that suggested that there is limited evidence to confirm the effectiveness of prohibiting mass gatherings [32]. In measuring the effectiveness of social distancing, Fong et al's systematic review evidence suggested that timely implementation and high compliance in the community would determine the success of the intervention [32]. The length of the intervention and compliance has also been identified to be an important factor in reducing the spread of respiratory infection in an epidemic situation [33], but there is limited evidence to confirm this.

Rashid et al, 2015 review reported local mobility restriction to have a peak delay effect, especially if implemented early into the pandemic, while a different study reveals that weak travel restrictions would lead to increased spread of the influenza virus [34]. It is therefore suggested that for this intervention to be effective, high restricted mobility should be implemented [34]. Jefferson et al's systematic review, which includes few studies from slums in developing countries, reports an impressive effect of hand hygiene in reducing respiratory transmission especially among younger children [18]. The review identified intervention compliance to be a problem despite the low-cost implementation [18], and encouraged incorporating simple

public health measures into structured programs like education to increases their affectivity in controlling the transmission of respiratory infections [18]. A systematic review on hand hygiene improvement strategies reported that studies revealed positive effect of hand hygiene strategy when a combination of determinants of behavior change, like knowledge, attitude, social influence, self-efficacy were applied. However, the best practice approach has not been determined [17].

Understanding the level of compliance on the community intervention to combat COVID-19 is critical in LMIC context to a several barriers and enablers in the settings. Therefore, this rapid review synthesizes evidence on the community measures and interventions available in LMICs and how compliance is encouraged in the community to inform Kenya's current efforts to combat the spread of COVID-19 in the country. In addition, the review will provide evidence on feasible mitigations from other outbreaks and pandemic experiences.

## Broad objective

To synthesize existing and emerging evidence on community interventions available in LMICs to inform COVID-19 control decisions within Kenya.

## Specific objectives

1. To understand barriers to, and enablers of, community measures and control of COVID-19.

2. To identify the best practices and innovative community measures in place to combat COVID-19.

3. To summarize the effectiveness and lessons learnt on community measures and control to combat previous epidemic diseases in the LMIC settings.

## Methodology

The rapid review method followed the Cochrane provisional rapid review methods recommendations. See link below for a rapid review recommended bare minimum sections to be included [35] https://methods.cochrane.org/rapidreviews/sites/methods.cochrane.org. rapidreviews/files/public/uploads/cochrane_rr_-_guidance-23mar2020-final.pdf. In addition, we used the PRISMA checklist to guide the team on the methodology [S1 Table]

## PICOST matrix

1. *Population*: Individuals of all ages located in LMICs.

2. *Intervention*: Community measures and control for infectious diseases. In this study community measures to be included are; hand hygiene, respiratory etiquette, use of masks and social distancing. The interventions may be at individual or community level.

3. **Study settings**: The variable of interest conducted in LMICs.

4. *Comparator*: Other community measure strategies to control an infectious disease.

5. **Outcome**: The outcomes of interest include:

   1. Implementation strategy on the mitigation measures in reference to COVID-19

   2. Completion and management of outbreak

   3. Coping mechanism by household on hygiene promotion and social distancing

6. ***Study design***: Reviews will include both interventional and observation studies as below;

- Intervention studies: individually randomized controlled trials (RCTs), cluster-randomized controlled trials (cRCT), non-randomized control trials, interrupted time series (ITS), and controlled before-and-after studies (CBAs).

- Observational studies: cohort studies, case-control studies, and cross-sectional studies. Case-studies will be included for emphasis purposes where applicable.

7. ***Time***: The timing of outcome assessment is similar to intervention timeline.

Note: For the purpose of this review, epidemic infectious disease outbreaks include Bovine Spongiform Encephalitis in 1986, the Avian flu in 1997, the SARS in 2002, the Swine Flu in 2009, and the Ebola in 2014.

## Exclusion criteria

The study had the following exclusions:

- Studies focusing on sick patients following confirmed cases

- Studies focusing on health care workers

- Modelling studies

## Search methods for identification of studies

We developed a comprehensive search strategy for peer-reviewed studies and grey literature with no time limit and language restricted to English studies only for the following databases: (a) PubMed https://www.ncbi.nlm.nih.gov/PubMed/, and (b) Google Scholar https://scholar.google.com/. The time limit was not restricted due to the reason that we had to summarize the lessons learnt on community measures and control to combat previous epidemic diseases that goes many years back up to 1986. Further, dedicated websites for emerging evidence on COVID-19 were searched including WHO https://www.who.int, MEDRXIV https://www.medrxiv.org/ and Google https://www.google.com/. We screened the MEDRXIV website with preprints on preliminary reports of work that have not been certified by peer review due to the nature of ongoing COVID-19 studies. In addition, the Google website gave us country specific news and government documents following the mitigation strategies in place. Five databases and/or websites were used due to the rapid nature of the study. We also screened the reference lists of all the included studies and related systematic reviews for other potentially eligible primary studies. Detailed search strategy is elaborated in S2 Table.

## Data screening and collection

According to the PICOST criteria above, due to the short turnaround of the review four authors (LA, JJ, CM & LM) independently screened through titles and abstracts of the retrieved records to identify potentially eligible studies. The full texts of the potentially eligible studies were also assessed using the pre-specified eligibility criteria. The four authors compared lists and discussed eligible studies and resolve any disagreements.

## Data extraction and analysis

A data collection form was designed, piloted and used independently by three review authors (LA, JJ & LM) to extract data from the included studies [S3 Table]. We extracted the following

information from each study: year, aims and purpose of the study, setting, type of disease, type of intervention, sample size.

Two authors (LA & JJ) conducted the analysis where we described quantitative data using standard summary statistics and performed meta-analysis using REVMAN software on the outcomes that meet the criteria for the rapid review. Where the outcomes of interest were related to face masking and hand hygiene interventions, we calculated risk ratios and their corresponding 95% confidence intervals and p-values for dichotomous outcomes, and mean differences and standard deviations for continuous outcomes. A random effects model was used with the assumption that the true effect size will vary between studies. For studies of similar interventions reporting similar outcomes statistical heterogeneity were examined using the chi-squared test for homogeneity (with significance defined at 10% alpha level). Statistical heterogeneity was quantified using the $I^2$ statistic >50%.

For the outcome measure related to social distancing intervention that was presented in qualitative format, analysis was based on thematic synthesis of qualitative research. Results of each included study were discussed using key descriptive themes such as demographics, study design and community interventions. The qualitative findings were independently coded by two authors (LA & JJ) with discussion within the study team to examine their relationship to the research questions and interpreted in reference to the research objective. Disagreements were resolved through discussion and a third author (LM) when the authors failed to reach consensus.

## Risk of bias (quality) assessment

The Cochrane Collaboration's risk of bias tool was used for cluster and individual randomized controlled trials and for non-randomized studies, the risk of bias in non-randomized studies of interventions (ROBINS-I) tool was used [36]. The risk of bias findings is elaborated in section 'risk of bias' below.

**Subgroup analysis.** For the subgroup analysis, the intervention/strategies, disease, study design and settings were considered during subgroup analysis of review data.

**Sensitivity analysis and assessment of reporting biases.** For the current review we planned to perform a sensitivity analysis based on risk of bias and missing data if we found sufficient data: however, available data were insufficient to perform this analysis. In addition, publication bias by intervention type were not assessed. The T test for asymmetry with a funnel plot was not feasible because the number of included studies for meta-analysis was less than the recommended 10 studies.

## Quality assessments

One author (JK) graded the certainty of evidence, with verification of all judgments (and noted rationales) by a second author (LA). The overall quality of evidence was conducted using a modified GRADE approach [37].

## Ethics approval

This is a rapid review and it did not require ethics approval.

## Results

### Results of the search

We identified 41,138 records from the electronic databases and other sources. After excluding 3,476 duplicates, we screened 37,662 records, and found that 37,602 records were not relevant to our review objective. We reviewed the remaining 60 potentially eligible full-text articles for

inclusion and excluded 43 of them with reasons listed below [Fig 1]. Seventeen studies met the inclusion criteria and were thus included [Tables 1 and 2]. Point to note is that some of the studies had more than one intervention in different arms i.e. face masking and hand hygiene. The search process and selection of studies is presented in the Prisma flow diagram below [Fig 1].

## Study design and setting

Seventeen studies met the inclusion criteria. Five studies reported on face mask [38–42]; six studies on hand hygiene [39, 40, 42–45]; nine studies reported on social distancing [46–53, 55]; and two studies reported on multi-component interventions [41, 45] i.e. combination of face mask and hand hygiene.

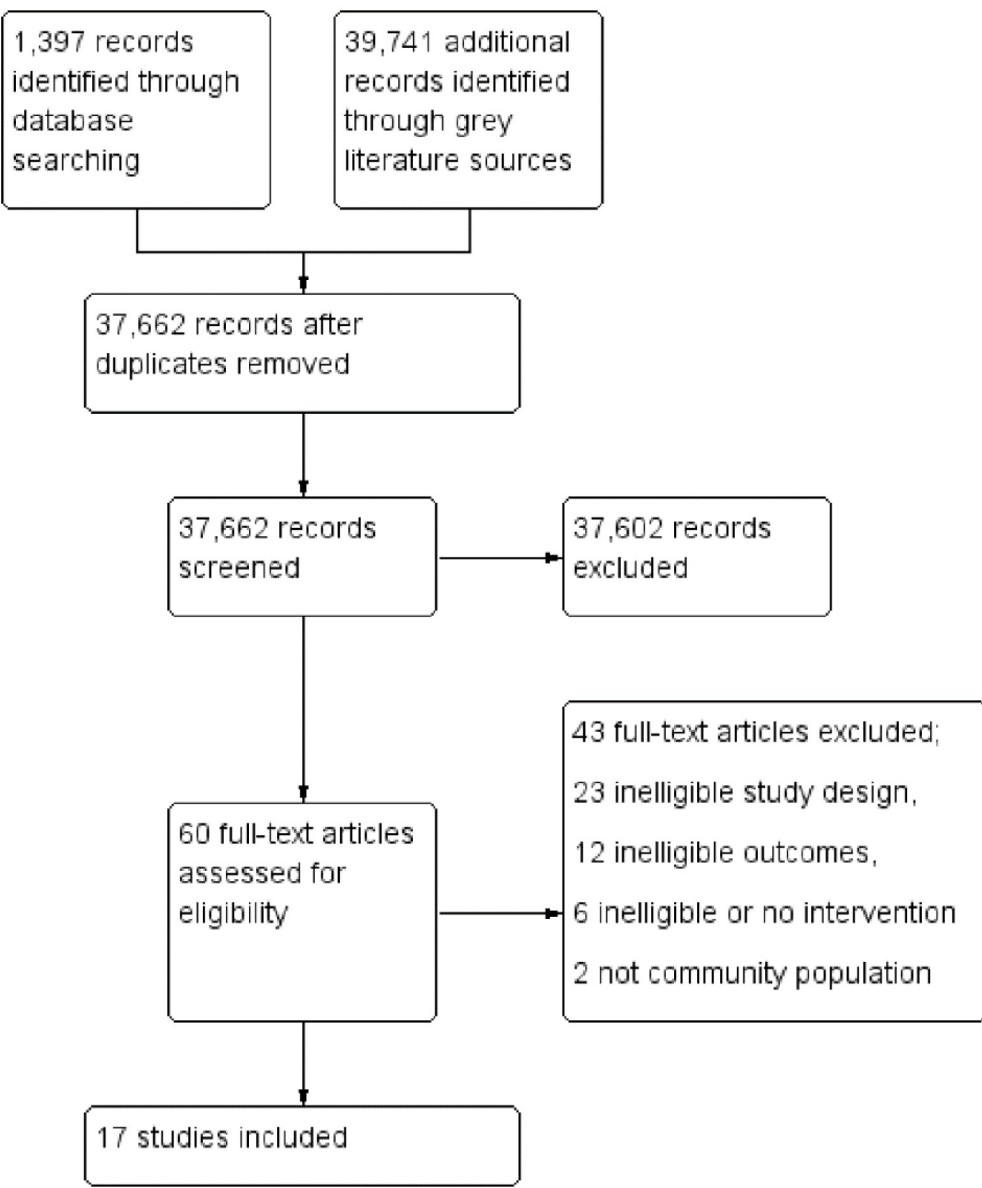

**Fig 1. Prisma flow diagram.** Fig 1 shows the process of selecting relevant studies from 41138 records. After removing 3476 duplicates, 37662 records were screened; 37602 of the records were excluded based on the title and abstract. Full texts of 60 potential eligible articles were retrieved and reviewed for inclusion. Of the 60 records, 17 studies met our inclusion criteria and 43 studies were excluded.

**Table 1. Characteristics of included studies: Face masking, hand hygiene and multicomponent intervention (face masking and hand hygiene).**

| Study ID | Country | Virus/ disease | Population | Study type | Outcome | Recommendations |
|---|---|---|---|---|---|---|
| *Face mask intervention* | | | | | | |
| Cowling et al 2008 [40] | China | Influenza | Household members | Cluster randomized trial | Study found little effect of the face masking intervention in preventing household transmission | Recommends more studies to provide evidence for the effectiveness of non-pharmaceutical interventions |
| Cowling et al 2009 [41] | China | Influenza | Household members | Cluster randomized trial | Results suggest that facemasks can reduce influenza virus transmission | Recommends use of facemasks and hand hygiene within 36 hours of index patient symptoms onset |
| Wu et al 2004 [39] | China | SARS | Community | Case-control study | Consistent mask use lowered the risk for disease | Recommends general community masking and other intervention to lower SARS and other respiratory infection |
| Zhang et al 2013 [38] | China | Influenza | Community | Case-control study | Wearing a face mask was associated with a decreased risk for influenza acquisition | Recommends a more comprehensive intervention study to accurately estimate the protective effect of face masks for preventing influenza virus transmission on long-distance flights. |
| Lau et al 2004 [42] | China | SARS | Household members | Case-control study | Face mask as a public health measure may have contributed substantially to the control of SARS epidemic | Recommends use of public Health measures in controlling respiratory epidemics |
| *Hand hygiene intervention* | | | | | | |
| Cowling et al 2008 [40] | China | Influenza | Household members | Cluster randomized trial | Study found little effect of the hand washing interventions in preventing household transmission | Recommends more studies to provide evidence for the effectiveness of non-pharmaceutical interventions |
| Ram PK et al 2015 [44] | Bangladesh | Influenza | Household members | Case-control study | Handwashing may reduce intra- and inter-household transmission of influenza | N/A |
| Simmerman et al 2011 [45] | Thailand | Influenza | Community | Randomized control trial | Influenza transmission was not reduced by intervention, Sociocultural factors had a role to play to improve future hand washing practices intervention | Recommends a prospective study design and a careful analysis of sociocultural factors that could improve future non pharmaceutical intervention studies. |
| Doshi et al 2015 [43] | Bangladesh | Influenza | Community | Case-control study | Study found no association between any of household handwashing measures and influenza infection since handwashing was practiced infrequently in the community | More robust research on interventions against influenza-specific risk factors to guide public health efforts in response to future influenza pandemics, when vaccines may not be readily available |
| Wu et al 2004 [39] | China | SARS | Community | Case-control study | Washing hands intermittently was associated with a smaller yet significant reduction in risk | Recommends general Community masking and other intervention to lower SARS and other respiratory infection |
| Lau et al 2004 [42] | China | SARS | Household members | Case-control study | Study shows that hand hygiene measures may have contributed substantially to the control of SARS epidemic | Recommends use of public Health measures in controlling respiratory epidemics |
| *Multi component intervention (facemask and hand hygiene vs control–intervention/hand hygiene only)* | | | | | | |
| Simmerman et al 2011 [45] | Thailand | Influenza | Community | Randomized control trial | Influenza transmission was not reduced by interventions involving promotion of hand washing and face-masking due to non-adherence | Recommends a prospective study design and a careful analysis of sociocultural factors could improve future non pharmaceutical intervention studies. |
| Cowling et al 2009 [41] | China | Influenza | Household members | Cluster randomized trial | Results suggest that hand hygiene and facemasks can reduce influenza virus transmission if implemented early after onset of symptoms | The study recommends use of facemasks and hand hygiene within 36 hours of index patient symptoms onset |

Table 1 shows a summary of the included studies for handwashing and face masking intervention i.e. study settings; disease investigated, study designs, target population and study outcome and recommendation.

For the quantitative studies on face masking and hand hygiene interventions, we had two studies that were cluster randomised trials with household as the unit of randomization [40, 41], one study was individual randomised control trial with interventions and control arm

**Table 2. Characteristics of included studies: Social distancing intervention.**

| Study ID | Disease | Country | Population target | Kind of social distancing | Duration of distancing | Effect of distancing | Timing distancing implemented | Recommendations |
|---|---|---|---|---|---|---|---|---|
| Flasche et al (2011) [48] | Influenza | Romania | General populations | School Holidays | Varied by country. | No evidence found of a relationship between infection control and the start of school holidays. | 2 weeks | Further research to enhance understanding of the precise mechanism behind distribution of susceptible cases and contact mechanisms to help predict the future spread of influenza more accurately and to design more efficient means to mitigate its impact. |
| Petrovic et al (2011) [50] | Influenza | Serbia | General population | School closure | 8 weeks | Disease rates declined following first closure and increased after schools reopened. | 4weeks | Recommends Severe Acute Respiratory illness surveillance and virologic surveillance in order to monitor the full scope of influenza pandemic |
| Chieochansin et al (2009) [46] | Influenza | Thailand | General population | Public holiday followed later by school closure | Public holiday occurred during peak week. Closure of schools followed | Incidence declined throughout the period of closure. | 2 weeks | Preventive measures to slow down the outbreak and thus enable health care centers to cope with the large number of respiratory tract disease. |
| Rajatonirina et al (2011) [51] | Influenza | Madagascar | Boarders at a school | School holiday | 2weeks | Epidemic appeared to be largely finished when the school closed. | 2weeks | A clear understanding of the spread of pandemic influenza A(H1N1) 2009 virus within a school setting and the impact of measures to interrupt transmission will help in preparing for future influenza virus pandemics. |
| Echevarria-Zuno et al (2009) [55] | Influenza | Mexico | National population | School closure | Approx. two weeks; entire education system For a week, then nationwide followed. | Epidemic was controlled during school closure | 2 weeks | N/A |
| Chowell et al (2011a) [47] | Influenza | Mexico | General population | School closure | ~7 weeks | Reactive closure appeared to slow epidemic growth, which resumed when interventions were lifted. | 8weeks and 4days | N/A |
| Chowell et al (2011b) [53] | Influenza | Peru | National population | School closure | 3 weeks, all schools nationwide | Cases decreased from peak week following closure | 2 weeks | N/A |
| Tinoco et al (2009) [52] | Influenza | Peru | General population | School closure | 3 weeks | Cases decreased throughout closure period | 2 weeks | Recommended more epidemiologic data on the impact of pandemic influenza from the Southern Hemisphere winter to help inform planning for the upcoming Northern Hemisphere influenza season. |
| Herrera-Valdez et al (2011) [49] | Influenza | Mexico | National population | One reactive closure and a subsequent school holiday | Reactive closure lasted ~2 weeks; holiday lasted ~2 months | Confirmed cases occurred in three waves corresponding to closing and reopening of schools. | 12 weeks | Availing more resources to increase the capacity of mass production of vaccines and treatment in preparation for a possibly more severe influenza epidemic in future |

Table 2 shows a summary of the included studies for social distancing intervention i.e. study settings; disease investigated, target population, duration, timing and effect of distancing and recommendations.

using individual as the unit of randomization [45]. and five studies were case-control studies [38, 39, 42–44]. Nine studies on social distancing intervention that were qualitative in nature were outbreak case reports [46–53, 55].

Five studies were conducted in China [38–42]; two studies were conducted in Bangladesh [43, 44]; three studies were conducted in Mexico [49, 53, 55]; two studies were conducted in Peru [52, 53]; two studies were conducted in Thailand [45, 46]; one study was conducted in Madagascar [51]; one study was conducted in Serbia [50]; and one study was conducted in Romania [48].

## Participants

One study enrolled children between 1yr and 5yrs residing in Bangladesh [44]; one study enrolled adolescents, hospital workers, inpatients, and residents/visitors [42]; 14 studies enrolled mixed participants comprising of children, adolescent and parents [39, 40, 41, 43, 45–54]; with one study enrolling passengers and crew team in a flight from New York to China, Hong Kong and from China, Hong Kong to Fuzhou [39].

## Excluded studies

We excluded 43 studies for various reasons included in the exclusion criteria. The most common reasons for exclusion were country not eligible for inclusion and ineligible interventions.

## Risk of bias in included studies

Three randomised control studies [40, 41, 45] were assessed using the Cochrane Risk of Bias tool 2.0 for randomised trials [54] while 14 non randomised studies were assessed using the risk of bias in non-randomized studies of interventions (ROBINS-I) tool [36]. Overall after consideration the risk of bias of randomized studies were generally moderate -to-high [Fig 2] while the risk of bias on the observational designs was generally low-to-moderate [Table 3].

On the quality of evidence, we graded the certainty of evidence using the GRADE approach. We used the GRADEpro app to rate evidence and presented in GRADE evidence profiles and summary of findings tables using standardized terms [37]. Quality grading for the 17 papers included in this review were classified as either very low, moderate, or high [S2 File].

## Interventions and comparisons

**Face mask interventions.**   The face mask intervention studies assessed the effectiveness of mask protection within case subjects and control subjects. Study by Zhang et al (2013) [38] compared the exposure of case-passengers with those of asymptomatic control-passengers with face masks, targeting the reduction in the infection rate of influenza. In another study [39], there was a dose-response effect where the persons who always wore masks had a 70% lower risk of being diagnosed with clinical SARS compared with those who never wore masks or intermittent mask use (60%). In two studies [40, 41], the control households group received education about the importance of a healthy diet and lifestyle, as an illness prevention while households in the face mask group received the control intervention plus education about the potential efficacy of masks in reducing disease spread to household contacts if all parties wear masks. Distribution of a box of 50 surgical masks for each household member, and demonstration of proper face-mask wearing and hygienic disposal was practiced in the intervention group.

**Hand hygiene interventions.**   The hand hygiene intervention studies assessed the efficacy of good hand hygiene within the case subjects and susceptible controls. In the study by Ram

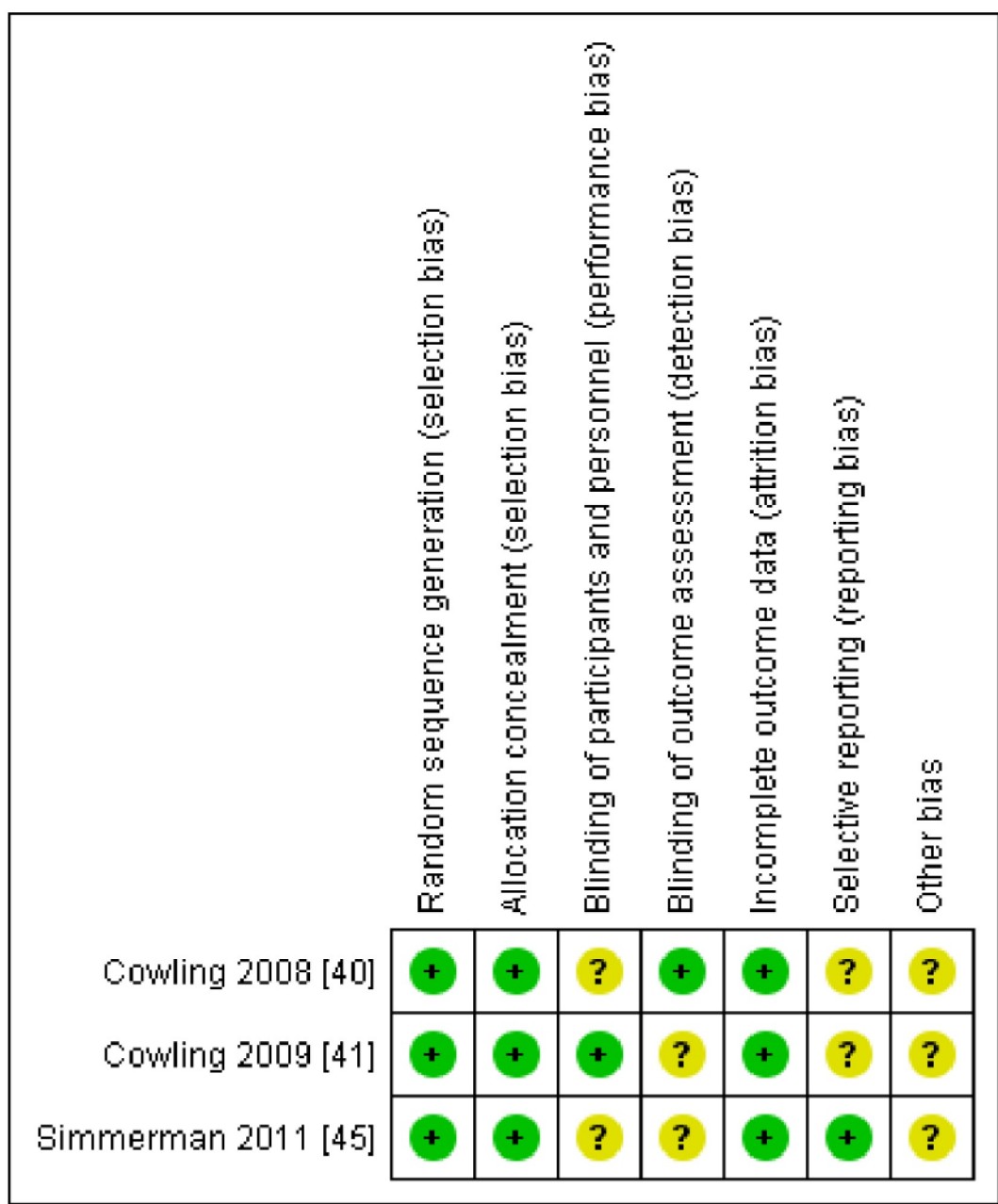

**Fig 2. Risk of bias summary.** Fig 2 shows the risk of bias graph for the randomised control trials studies.

et al (2015) where the Influenza infection was confirmed in 20% of controls and 12% of the intervention group, hand hygiene intervention (daily handwashing with soap promotions) aimed at reducing the infection rate of disease [44]. In another study [40], the control households group received education about the importance of a healthy diet and lifestyle while households in the hand hygiene group received the control intervention plus education about the potential efficacy of proper hand hygiene in reducing transmission, distribution of an automatic alcohol hand, liquid hand soap, individual small bottles of alcohol hand gel, and demonstration of proper hand washing and hand anti-sepsis. A study by Doshi et al [43] observed

**Table 3. ROBINS-I risk of bias summary: Review authors' judgements about each risk of bias domain for each included study.**

| Studies | Bias due to confounding | Bias in selection of participants into the study | Bias in measurement of interventions | Bias due to departures from intended interventions | Bias due to missing data | Bias in measurement of outcomes | Bias in selection of the reported result |
|---|---|---|---|---|---|---|---|
| Wu et al 2004 [39] | Y | N | N | U | U | Y | N |
| Zhang et al 2013 [38] | Y | Y | N | N | N | N | N |
| Lau et al 2004 [42] | N | Y | N | Y | N | Y | N |
| Ram PK et al 2015 [44] | Y | N | N | N | Y | N | Y |
| Doshi et al 2015 [43] | N | N | N | N | N | N | N |
| Flasche et al (2011) [48] | U | U | U | U | U | U | U |
| Petrovic et al (2011) [50] | U | U | U | U | U | U | U |
| Chieochansin et al (2009) [46] | U | U | U | U | U | U | U |
| Rajatonirina et al (2011) [51] | U | U | U | U | U | U | U |
| Echevarria-Zuno et al (2009) [55] | U | U | U | U | U | U | U |
| Chowell et al (2011a) [47] | U | U | U | U | U | U | U |
| Tinoco et al (2009) [52] | U | U | U | U | U | U | U |
| Chowell et al (2011b) [53] | U | U | U | U | U | U | U |
| Herrera-Valdez et al (2011) [49] | U | U | U | U | U | U | U |

Table 3 shows assessment of risk of bias for the observational studies

household handwashing behaviour. Soap consumption was estimated by summing weight differences of three bars of soap sequentially left in each household. One study by Wu et al observing the hand washing practices behaviour among the intervention and control participants found that consistently washing hands upon returning home was associated with a reduced risk for clinical SARS [39]. Study by Simmerman et al the where households were randomized to control, hand washing (HW), or hand washing plus paper surgical face masks (HW + FM) arms [45]. One matched case control study was conducted in Hong Kong followed up on the frequent hand washing, and disinfecting the living quarters among the control and interventions group [42].

**Social distancing interventions.**   The social distancing intervention studies assessed the importance of physical distance through closure of learning institutions on the transmission dynamics of disease among the case subjects and the control individuals [46–53, 55]. The intervention included school closures, closure of public spaces, and the cancellation of public events [46–53, 55]. All the studies [47–53, 55] except one [46] were consistent to show the class dismissal policy in schools as one of the effective pandemic mitigation strategies. One of the studies in Thailand observed that school closures to slow down the spread of influenza did not lead to a significant reduction in influenza-like illness when compared to the rate of influenza-like illness in areas where no holiday had been introduced after the peak and class dismissal policy [46]. There were no studies that have published the effect of social distancing on COVID-19 in LMICs.

**Multi-component interventions.**   Two studies assessed the multi-component interventions which involved use of both face masks and good hand hygiene within the case subjects and control subjects in the study aimed at reducing the infection rate of disease [41, 45]. One study [41] had three arms involving two interventions arms and one control arm. Households in group 1 received hand-washing education and a hand-washing kit that included a graduated dispenser with standard unscented liquid hand soap; Households in group 2 received hand-washing education and the hand-washing kit, and a box of 50 standard paper surgical face masks and 20 pediatric face masks while the control group received nutritional, physical activity, and smoking cessation education. In a study by Simmerman et al, the households were randomized to control, hand washing (HW), or hand washing plus paper surgical face masks (HW + FM) arms [45].

## Outcomes following community measures

The outcomes following community measures are discussed broadly under four sub-sections:

1. Implementation strategy on the mitigation measures in reference to COVID-19

2. Successful management of an outbreak disease

3. Coping mechanisms by communities and households on hand hygiene and social distancing

4. Barriers to, and enablers of, community measures and control of COVID-19

It's worthwhile to mention that except the result obtained on the section 'successful management of an outbreak' the other subsections obtained information available from media stories that is neither scientific nor systematic hence the quality of evidence is not clearly stated in individual references. The quality of evidence on studies discussed in the section 'successful management of an outbreak' is elaborated using summary of finding table in S2 File.

**a) Implementation strategy on the mitigation measures in reference to COVID-19.** COVID-19 is a rapidly emerging disease with close to 3 months since it was classified a pandemic by WHO. In the absence of treatment for the virus, there are a few grey materials in LMIC that have documented the public health strategies to control the spread. However, the interventions are a work in progress as their efficacy is yet to be proven. Below is a summary of the strategies documented from grey unpublished literature.

*Individual, community and environmental measures.* These are individual/community based preventive measures that include hand hygiene practices, wearing of facemasks and maintaining clean environment, including water and sanitation. Currently, most of the countries are having daily briefings and using social media platforms to enhance knowledge, attitude and practice on hand hygiene and use of masks. On adherence to community intervention, Its important to ensure availability of supplies including masks, sanitizers, soap and water in all settings, including public spaces, refugee settings, and informal settlements.

*Detecting cases.* As the COVID-19 pandemic escalates, "test, test, test" has been the tune for defeating the novel coronavirus with the intention of positive cases to either be isolated in a facility or put on strict home isolation which is a challenge in LMIC setting. However, most of the LMICs are finding themselves at the end of a long global queue for the polymerase chain reaction (PCR) chemical reagents and other commodities necessary for administering diagnostic tests. So far, South Africa and Ghana have accounted for nearly half of all tests carried out on the continent with over 500,000 and 115,000 tests carried out in South Africa and Ghana, respectively [56]. The WHO country offices in LMICs have supported countries to rapidly build and scale up the testing capacity for COVID-19. However, there is still a scarcity of

testing kits and reagents in many countries. One strategy is to develop successful rapid diagnostic test kits (RDTs) in LMIC settings to ensure local availability.

*Contact tracing and quarantine*. Contact tracing and isolation or quarantine of sick or exposed individuals are among the most effective tools to reduce transmission of infectious disease. Closely watching these contacts after exposure to an infected person helps the contacts to get care and treatment, and prevents further transmission of the virus. Governments play a big role in contact tracing where there is a 24/7 workforce for contact tracing and there is a dedicated public and private space for quarantine. However, governments are struggling to trace some hundreds of people who have tested positive because they gave wrong contact details due to the phobia of being taken to a quarantine facility. Currently, quarantine on average lasts for 14 days, however, there are scientific reports showing that the incubation period of COVID-19 could go way beyond 14 days.

*Social and physical distancing measures*. Social and physical distancing is the most effective, but the most challenging intervention to measure so far. Social and physical distancing measures implemented so far include banning mass gatherings, school closures, and restriction of local and international travel. Evidence from unpublished material document that implementation of social and physical distancing can curb the pandemic by 95% [57]. Though the benefit is clear, most countries are experiencing challenges with the coordination and implementation of social distancing, as to when, into the pandemic is the best time to implement the measure.

**b) Successful management of an outbreak disease.** *Face mask intervention in management of an outbreak disease*. In a subgroup analysis, 5 studies [38–42] reporting on the effectiveness of wearing masks included 2,717 participants. In general, masks are effective in preventing the spread of Influenza and SARS viruses among the general population. After wearing a mask, the risk of contracting SARS and Influenza was reduced, hence protective effect, with the pooled OR 0.78 and 95% CI = 0.36–1.67 ($I^2$ = 16%, M-H Random-effect model) [Fig 3]. We judged the certainty of the evidence as moderate because of study limitations, as the included study had high risk of bias and imprecision of findings with a wide confidence interval in the included study [Table A in S2 File].

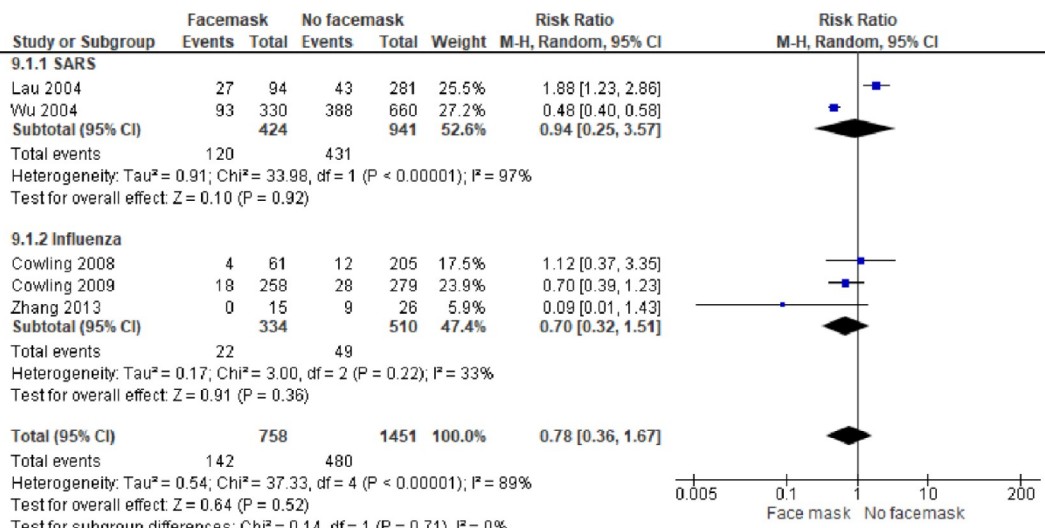

**Fig 3. Face mask intervention only in management of an outbreak disease.** Fig 3 shows the effect of face mask intervention in preventing influenza and SARS viruses among population.

*Hand hygiene intervention in management of an outbreak disease*. In a subgroup analysis, 6 studies [39, 40, 42–45] reporting on the effectiveness of hand washing included 3,665 participants. In general, hand hygiene is effective in preventing the spread of influenza and SARS viruses among the general population. Following hand hygiene practices, the risk of contracting SARS and Influenza was reduced slightly, hence protective effect, with the pooled OR 0.95 and 95% CI = 0.83–1.08 ($I^2$ = 0%, M-H Random-effect model) [Fig 4]. We judged the certainty of the evidence as low and moderate respectively for SARS and Influenza because of study limitations, as the included study had high risk of bias and imprecision of findings with a wide confidence interval in the included study [Table B in S2 File].

**Multi-component interventions.** *Face mask and hand hygiene vs hand hygiene only*. Two studies [41, 45] that included 1,679 participants compared the effectiveness of combined intervention of wearing face masks and hand hygiene versus hand hygiene only as control. Compared to hand hygiene only, the combined intervention of face masks and hand hygiene did not show effectiveness in preventing the spread of influenza among the general population. One study [45] reported risk of contracting Influenza was increased slightly in the group practicing both face-masking and hand hygiene due to non-adherence on using the intervention leading to the pooled result of no effect on the combined intervention. Hand hygiene only had a protective effect, with the pooled OR 1.09 and 95% CI = 0.78–1.50 ($I^2$ = 0%, M-H Random-effect model) [Fig 5]. We judged the certainty of the evidence as high because of study limitations, as the included study had unclear risk of bias in the included study [Table C in S2 File].

*Face mask and hand hygiene vs control*. The same two studies [41, 45] compared the effect of the combined face mask and hand hygiene intervention versus no intervention (control group). The combined face mask and hand hygiene intervention showed effectiveness in controlling the transmission of influenza compared to the control group (no intervention). Following the wearing of a mask and hand hygiene, the risk of contracting Influenza was reduced, hence protective measure, with the pooled OR 0.94 and 95% CI = 0.58–1.54 ($I^2$ = 56%, M-H Random-effect model) [Fig 6]. We judged the certainty of the evidence as high because of study limitations, as the included study had unclear risk of bias [Table D in S2 File].

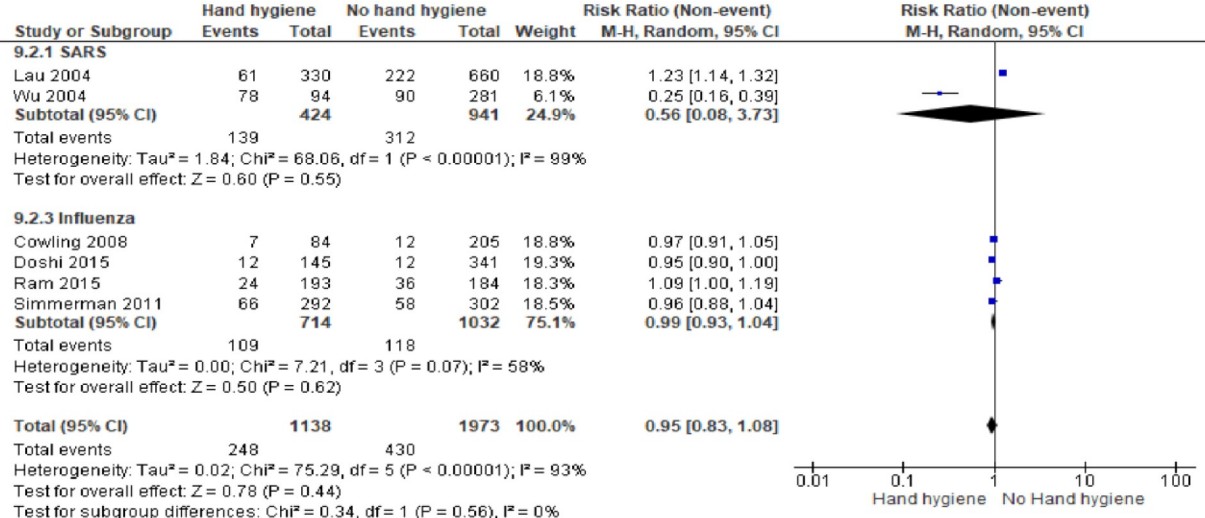

**Fig 4. Hand hygiene intervention only in management of an outbreak disease.** Fig 4 shows the effect of hand hygiene intervention in preventing influenza and SARS viruses among population.

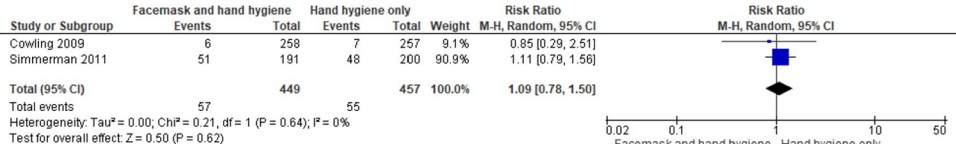

**Fig 5. Face mask and hand hygiene vs hand hygiene only.** Fig 5 shows the no-effect of face mask and hand hygiene intervention compared to hand hygiene only in preventing influenza among population.

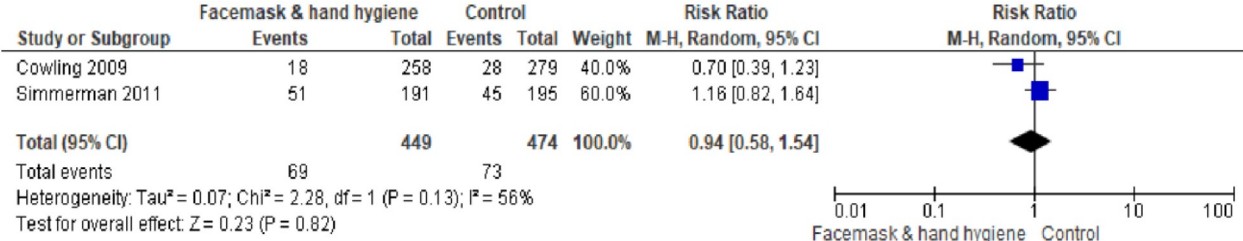

**Fig 6. Face mask and hand hygiene versus control group (no intervention).** Fig 6 shows the effect of face mask and hand hygiene intervention compared to no intervention in preventing influenza among population.

**Social distancing intervention for management of outbreak disease.** Nine studies assessed the social distancing intervention through physical distance. The intervention included school closures, closure of public spaces, and the cancellation of public events [46–53, 55]. Minimizing gathering i.e. school closures was found to slow down the spread of influenza. We judged the certainty of the evidence as low because of imprecision of findings, unclear risk of bias & high risk of bias [Table E in S2 File].

**c) Coping mechanisms by communities and households on hand hygiene and social distancing.** *Hand hygiene*. Most households in Kenya do not have running tap water, including majority of households in rural areas where 73% of the Kenyan population lives [58, 59] and 56% of Kenyan urban dwellers who reside in slums [60], and those living in refugee camps. To practice hand hygiene, innovative Kenyans have come up with ways of ensuring they have running tap water. For instance, in Kibera slums in Nairobi, where frequent hand hygiene is almost next to impossible, a local community leader has worked with residents to construct hand washing stations using locally available material, distribute soap and dig wells for water provision throughout. The hand washing stations also serve as educational centers where volunteers share information on the importance of hand hygiene in combating COVID-19.

Still in Kenya, matatu (private minibus) drivers are required by the government to: provide hand sanitizers for passengers upon boarding, clean vehicles twice per day, and keep detailed list of passengers.

Preparation of hand sanitizers using cheap, easily available, but safe products has been embraced by most developing country slum dwellers [61]. Community leaders have also come up with ways of ensuring residents are supplied with hand sanitizers through bulk purchasing and dividing in smaller portions and through promotion of small community projects [62].

In Ethiopia, a young inventor has developed a contact-free hand sanitizer dispenser in order to promote hand hygiene and reduce surface contact. His invention has been embraced by the local community and patented by the SaveIdeas organization. Through his idea, 50 dispensers have been produced and distributed in different hospitals [63].

*Social distancing*. Governments around the world have directed their citizens to adhere to social distancing guidelines provided by WHO [64] in order to limit the spread of the coronavirus. Globally, the type of social distancing measures implemented includes complete and/or partial lockdowns, shifting to remote working, and online schooling due to school closure. However, in LMICs these guidelines may not be practical, and are in some contexts poorly understood and/or weakly enforced. WHO recommended social distancing measures may not be practical in LMIC settings due to social networks, small and often crowded informal/slum settlements, and large families with minimal house space that make social distancing impossible to maintain. Regardless of the difficulties to implement and adapt the social distancing measures, many LMICs have adopted lockdown policies where majority of citizens are advised to stay home. Some of the social distancing adaptations include:

1. Social distancing through minimal/partial movement through border restriction/cessation

   - Border closures and travel restrictions (suspended flights or airport closures).

   - Imposing of dusk to dawn curfews, partial lockdowns or full lock downs (Kenya and Senegal) [65–69]

   - Risk-based movement restrictions rather than blanket restrictions across the country. For instance, in Kenya, movement restrictions have been imposed in transmission hotspots—Nairobi, and Coastal Counties, and some residential neighborhoods in Nairobi and Mombasa, rather than the entire country (Kenya, Ghana, Nigeria)

2. Social distancing through school closures

   - Nearly all countries have temporarily closed schools and non-essential businesses or banned social gatherings [70]

3. Social distancing in providing/obtaining essential services

   - Allowing food markets and small-scale traders to operate with measures to reduce physical distance such as reducing the number of traders and customers, relocating traders to decongest markets, and hygiene (Kenya, South Africa) [65]

   - Opening markets on specific days and times of the week, and closing them on other days and times. For instance, in Nigeria and South Africa, markets are open on specific days of the week, and for a shorter time on the open days [65]

   - For essential services, require social-distancing when stores remain open. Shopkeepers, vendors, and all facilities welcoming the public can use objects (stones, cans) or draw lines with chalk or paint to indicate how people should queue to encourage these practices

   - Request restaurants, bars, and snack- and tea/coffee vendors–even the most informal ones–not allow people to sit down or congregate around their establishments, and only allow take-away

4. Social distancing in public and other transport

   - Banning public transportation that are suspected to be particularly risky: Rwanda for instance banned moto-taxis [71] and Uganda banned movement of both public and private motors [65]

   - Limit the number of passengers: The DRC, Senegal, Kenya and Rwanda have limited the number of passengers allowed to board buses [65]

*Face mask interventions*. Requiring the people to wear (cloth-based) face masks in public spaces [72]. Makeshift cloth masks are encouraged to cushion communities against the face mask expenses.

*Multi-component interventions*. Requiring shopkeepers and vendors to practice excellent hand hygiene and wear a cloth over their noses and mouths as well as gloves. Vendors should wash hands with soap and water before and after each transaction, or use hand sanitizer in its place if running water is not available.

### d) Barriers to, and enablers of, community measures and control of COVID-19

Currently with COVID-19, unpublished evidence has shown that there are several factors in LMIC contexts that make it difficult to design and implement extensive community measures. The [Fig 7] below summarizes the current barriers and enablers to achieve implementation of the community interventions to mitigate the spread of COVID-19 in Kenya. The barriers can act as an opportunity for the government of Kenya to enhance the uptake and compliance of interventions.

### Effects of interventions

**Health system impact.** The health system is normally heavily burdened during outbreaks especially in LMICs where health systems are already strained. Health policy measures such as working from home, travel restrictions, the closure of schools, the ban on public gatherings and curfews, mandatory use of face masks, and frequent hand hygiene are necessary to reduce the spread respiratory infections [32]. Staying at home as a strategy to minimize the spread of COVID- has resulted in non-use of immunization and other health care services. WHO,

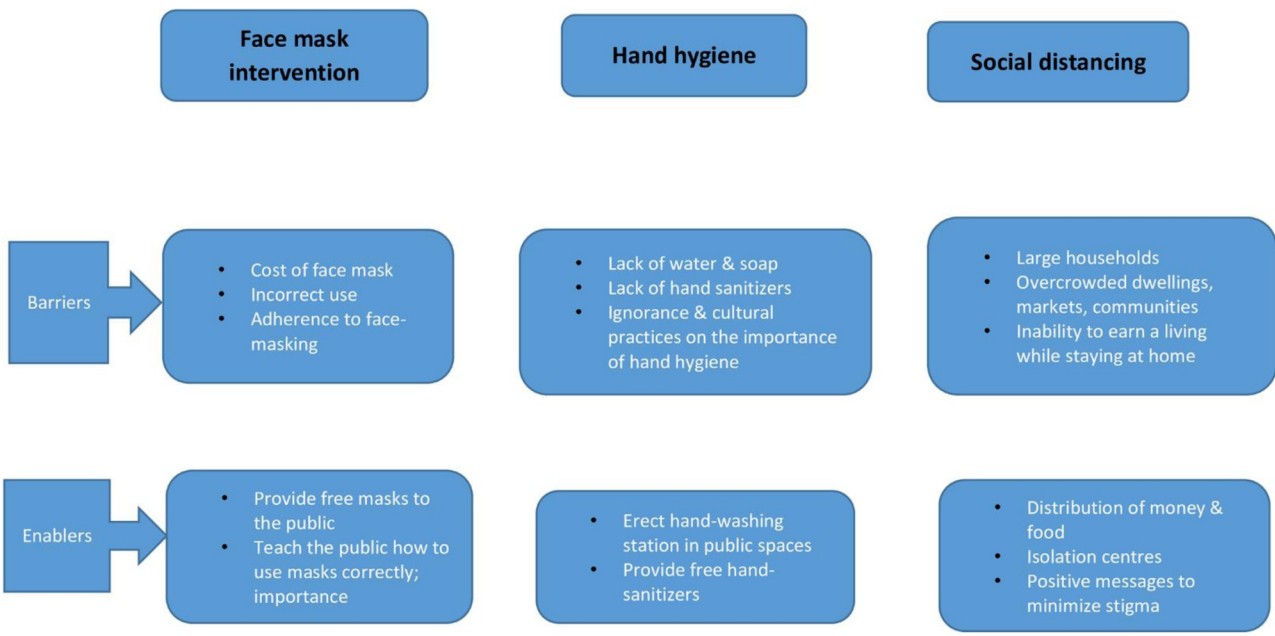

**Fig 7. Barriers to, and enablers of, community measures and control of COVID-19.** Fig 7 shows the summary on the current barriers and enablers to achieve implementation of the community interventions to mitigate the spread of COVID-19.

UNICEF and GAVI have reported that at least 80 million children under the age of one year in both developed and developing countries are at risk of diseases such as diphtheria, measles and polio [73] because they are failing to get immunized during the COVID-19 pandemic.

Hand hygiene and respiratory etiquette has proven to ease the burden of infections such as diarrhea and influenza, to the health-care system. Kirsch et al reported decline in the epidemic curve due to community intervention measures that were put in place during the Ebola virus epidemic in Liberia, and recommended the strengthening of behavioural change and burial practices [74]. A systematic review by Jefferson et al also points to the tremendous effect of hand hygiene especially among young children [18] indicating some ease effect to the health-care system.

Despite the positive impact on use of face mask in infection reduction [25, 27], the mandatory use of face masks has led to shortage of N95 masks all over the world, which is a necessity when handling COVID-19 patients [75]. Nurses and doctors from different parts of the world have shared their predicaments on the dangers they face as they are expected to save lives and at the same time stay safe. These has seen several healthcare workers contracting the infection, which has further put a strain on health-care system [75].

**Social-economic impact.** Non-pharmaceutical interventions implemented during outbreaks have been shown to yield positive results in containing the infection [32, 40, 76] and behavioral change [74]. On the other hand, these measures have caused other effects in equal measure. For instance, social distancing through total school closure has led to adverse effects including economic harm to working parents who are forced to stay at home and care of the children, thereby losing daily income [77]. This has also affected children whose nutritional source is mainly from the meals offered at school leading to nutritional problems [78]. With the implementation of the dusk to dawn curfews and stay at home policy, the 'new normal' has led to solitude adjustment and socialization has become a thing of the past with individuals retreating into their houses without any neighborhood checks. These can lead to psychological problems and such have been reported [77].

Quarantine as a way of social distancing individuals suspected to have contracted the disease or case contacts has been shown to cause post-traumatic stress. A study comparing post-traumatic stress in children and parents who were quarantined and those not quarantined found the measure was higher in the quarantined category than those who were not quarantined [79]. These mitigations have also had extreme financial effects [80]. As a result of low business profit, employers have opted to implement salary reductions, contract terminations, and forced unpaid leave in order to keep their businesses afloat. Globalization and interconnectedness has also been affected by the measures. The stay at home policy and total border closure has resulted in minimal movement of goods and people, affecting business operations. Inability to sufficiently import goods from other countries has led to hiked prices of goods and services. It is estimated that imports from China into Kenya to date have reduced by over 36% [81].

**Education impact.** The education sector has mainly been affected by the social distancing policy, which requires all children to stay at home following school closure policy implementation [77]. The greatest effect to the education system is the loss of education time which disrupts different curriculum programs [77]. It also leads to delayed internal and national assessments, which is viewed with a lot of uncertainties [82]. Most institutions are trying to solve the problem by introducing online programs; however, this also introduces an advantage imbalance among the students; in Africa, only 24% of the population can access Internet, which majority of students are not able to join online learning programs. Online learning programs are also severely affected by frequent power outages [83].

## Discussion

In the absence of a cure or a vaccine for COVID-19, countries have to rely on non-pharmaceutical interventions to reduce the spread of the disease, including: testing, contact tracing, isolation and treatment, hand hygiene, face masks, and a range of physical distancing measures. Our rapid review synthesized existing and emerging evidence on community interventions implemented in LMICs to inform COVID-19 control measures in Kenya. The study documents the effectiveness of the community measures, the barriers to, and enablers of, the community measures, innovative and best practices by communities in implementing and/or adapting COVID-19 control measures.

Our review was comprehensive as we included all known types of interventions for reducing the transmission of COVID-19 infection in the community including social distancing, hand hygiene, and face masking. We did not find any research/evidence/studies on community interventions on COVID-19 in LMICs. So we focused on evidence/studies on previous outbreaks i.e. SARS and Influenza to document the effectiveness of community interventions. We identified 17 eligible studies in LMICs. Over and above, we looked at unpublished websites that have evidence under peer review that capture the ongoing evidence on COVID-19 in the target setting. The unpublished materials reported the best practices and policy that lead to social distancing, hand hygiene and face mask implementation in current COVID-19 situation.

Worldwide, people are staying home to reduce transmission of the SAR-CoV-2, the virus that causes COVID-19. On social distancing, school closure and work place arrangement related interventions are the most researched type in non COVID-19 outbreaks i.e. influenza, SARs and Ebola with handful of studies from LMICs. Evidence shows effective measures like school closure and work place arrangement have reduced the transmission of Influenza, SARS and related outbreaks in delaying the peak of an epidemic. On the other hand, there are no published studies on COVID-19 on social distancing, hand hygiene and use of face masks in LMICs, rather we have unpublished studies with ongoing evidence.

On the intervention on face masks, there are contested discussions on whether masks will reduce transmission of COVID-19 in the general public [84, 85]. However, WHO acknowledges that the wearing of masks by the general public has been impactful in reducing previous severe pandemics [86]. Evidence has shown that even partial protective effect could have a major influence on transmission [87]. Hence, in many contexts, the proper guideline on use of face masks is missing in the community. Mishandling and inappropriate use of masks have been shown in the grey literature. Most of the LMIC communities are advised on the use of makeshift cloth masks and hence sensitization on the use and handling of face masks is critical. There are existing global guidelines on the use of face mask among other interventions, e.g. cloth face coverings should not be placed on children under the age of two years. Therefore, it is advisable for LMICs to adapt the recommended global guidelines to their contexts to enhance practical understanding on the use of face masks.

A number of studies have documented the potential of hand hygiene interventions for reducing previous outbreak infections in the community. However, the evidence does not distinguish between hand washing with soap or hand sanitizer. It is clear that use of soap and/or hand sanitizer use have different resource implications and are differentially effective in eliminating certain pathogens. Additionally, many sub-populations in LMICs lack safe drinking water let alone the water for cleaning hands and so the recommended advice on frequent hand washing practices is difficult to heed. The need might lead to small-scale solutions like setting up a network of public hand-washing stations as most of the countries are doing. The setting up of hand washing stations in public places should be delegated to certain department within the government to enhance accountability and efficiency.

Studies on the multi-component interventions had mixed findings i.e. combination of one or more interventions is more efficient than no intervention while combination of interventions has slightly no improvement compared to a single intervention i.e. hand hygiene only. The slightly no improvement was linked to non-adherence to interventions. Overall, the findings of this rapid review align with other reviews emphasizing the value of multi-component interventions. Based on the evidence, it is advisable to put in place multi-component community measures that combine social distancing with use of face mask and hand hygiene to save lives.

In a number of LMICs, communities have been shown to lack knowledge and guidelines on proper use of face masks, this has resulted in mishandling and inappropriate use of masks. Either individuals are not aware of the guidelines, do not understand the specific steps to follow, or they are not convinced of the need to practice these behaviors. This points to the need for continuous sharing of information to increase the public's awareness about the pandemic, its risks and prevention measures that has been shown to be effective in improving adherence. One proposed measure of improving adherence would be incorporating simple public health measures into structured programs such as the national education program, so as to increase their effectivity in controlling the transmission of respiratory infections.

In cases where non-adherence is resulting from unavailability of preventive health products such as masks, lack of water and hygiene materials, the government needs to intervene to provide the recommended preventive products/materials, at no cost, where possible. This can help ensure improved uptake. In addition, there is need for government to implement strict measures to enforce compliance in order to increase adherence to the community-level control measures. The review highlights the gaps in studies that show effects of strictly adhering to social distancing, hand hygiene and use of face masks to mitigate the spread of COVID-19.

## Conclusions

In Kenya like all other LMICs, adherence to the intervention has been cited to be the biggest challenge in the war against COVID 19. Adherence strategy is key in order to ensure that the mitigation measures will positively contribute to flattening the infection curve. Adherence to mitigation measures has been shown to be influenced by substantive moral support and social norms [88], nevertheless this would be dependent on the population setting within LMIC context. Adherence could be encouraged through public health education on the dangers of COVID 19 [18], and use of behavior change messages to encourage preventive practices. Understanding the sociocultural practices relevant to different communities is key in order to rightfully implement the measures and encourage compliance [45, 89] for example patient care norms, in cases of isolation and quarantine. Continuous surveillance and research on effectiveness of the mitigation strategy rolled out by the government is important in informing and evaluating policy.

### Implications for practice

Effective recommended interventions i.e. social distancing, hand hygiene and wearing of face masks in LMICs requires creativity and adaptation to local contexts, which could vary dramatically across regions and precisely in Kenya. The emerging and ongoing evidence on best practices globally on combating the COVID-19 is evident in progress and needs to be tracked frequently to help to inform response efforts, amended and adjusted for local needs.

### Implications for research

There are numerous limitations on the evidence highlighting the difficulties of conducting research on this topic in the community setting for both experimental and observational

designs. For example, hand hygiene is a non-invasive, non-pharmaceutical intervention without adequate comparator interventions. There are also challenges in conducting RCTs with appropriate sample sizes to establish effectiveness of hand hygiene, social distancing and use of face masks. In the community setting, it is also difficult to implement interventions and assess outcomes. Therefore, in light of the robust body of evidence on the benefits of community interventions to reduce disease spread, the compelling evidence is strong on the use of various recommended interventions to reduce the risk of infection and transmission in the community.

## Supporting information

**S1 Table. PRISMA guideline.** S1 Table shows the Preferred Reporting Items for Systematic Reviews and Meta-Analyses.
(DOCX)

**S2 Table. Search term in PubMed.** S2 Table shows the detailed search term used in PubMed in the study.
(DOCX)

**S3 Table. Data extraction form.** S3 Table shows the data extraction form template used in the study.
(DOCX)

**S1 File. Minimum data used in the analysis.**
(DOCX)

**S2 File. Summary of finding table.** S2 File shows the Summary of Finding table on the quality of evidence on the included 17 studies.
(DOCX)

## Acknowledgments

The study was a collaboration among partners of the Heightening Institutional Capacity for Government Use of Health Research (HIGH-Res) in Kenya i.e. African Institute of Development Policy (AFIDEP), Ministry of Health (MoH), and Kenya Medical Research Institute (KEMRI).

## Author Contributions

**Conceptualization:** Leila Abdullahi, Carol Mukiira, Joyce Wamicwe, Rachel Githiomi, David Kariuki, Peter Wanjohi, George Githuka, Charles Nzioka, Jennifer Orwa, Rose Oronje, James Kariuki, Lilian Mayieka.

**Formal analysis:** Leila Abdullahi, John Joseph Onyango, Carol Mukiira, Lilian Mayieka.

**Methodology:** Leila Abdullahi, John Joseph Onyango, Carol Mukiira, Lilian Mayieka.

**Project administration:** George Githuka.

**Validation:** Cosmas Mugambi.

**Writing – original draft:** Leila Abdullahi, John Joseph Onyango, Carol Mukiira, Joyce Wamicwe, Rachel Githiomi, David Kariuki, Cosmas Mugambi, Peter Wanjohi, George Githuka, Charles Nzioka, Jennifer Orwa, Rose Oronje, James Kariuki, Lilian Mayieka.

**Writing – review & editing:** Leila Abdullahi, Joyce Wamicwe, Rachel Githiomi, David Kariuki, Peter Wanjohi, George Githuka, Charles Nzioka, Jennifer Orwa, Rose Oronje, James Kariuki, Lilian Mayieka.

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
