## [Decision Letter · Decision Letter 0]

14 Sep 2020

PONE-D-20-21265

Community Interventions in Low- and Middle- Income Countries to Inform COVID-19 Control Implementation Decisions in Kenya: A Rapid Systematic Review

PLOS ONE

Dear Dr. Leila H Abdullahi,

Thank you for submitting your manuscript to PLOS ONE. After careful consideration, we feel that it has merit but does not fully meet PLOS ONE’s publication criteria as it currently stands. Therefore, we invite you to submit a revised version of the manuscript that addresses the points raised during the review process.

We look forward to receiving your revised manuscript.

Kind regards,

Francesco Di Gennaro

Academic Editor

PLOS ONE

Journal Requirements:

2. We note that the results of your quality assessment using the Cochran and Robins scales are not shown. Please include a table showing how each study scored on each item of these scales. moreover, please conduct a publication bias assessment.

3.We suggest you thoroughly copyedit your manuscript for language usage, spelling, and grammar. If you do not know anyone who can help you do this, you may wish to consider employing a professional scientific editing service.  

4.In your Data Availability statement, you have not specified where the minimal data set underlying the results described in your manuscript can be found. PLOS defines a study's minimal data set as the underlying data used to reach the conclusions drawn in the manuscript and any additional data required to replicate the reported study findings in their entirety. All PLOS journals require that the minimal data set be made fully available. For more information about our data policy, please see http://journals.plos.org/plosone/s/data-availability.

5.Thank you for stating the following in the Acknowledgments Section of your manuscript:

[This review was resourced through the Heightening Institutional Capacity for Government Use of

Health Research (HIGH-Res) project, which is funded the Alliance for Health Policy and Systems

Research (AHPSR).]

 [The funders had no role in study design, data collection and analysis, decision to publish, or preparation of the manuscript.]

8. We note you have included a table to which you do not refer in the text of your manuscript. Please ensure that you refer to Table 2 in your text; if accepted, production will need this reference to link the reader to the Table.

9. Please ensure that you refer to Figure 8 in your text as, if accepted, production will need this reference to link the reader to the figure.

10.We note that Figure [6 and 7] includes an image of a patient / participant in the study. 

Additional Editor Comments (if provided):

Dear authors,

follow reviewer suggestions to improve your article

Reviewers' comments:

Reviewer's Responses to Questions

**Comments to the Author**

1. Is the manuscript technically sound, and do the data support the conclusions?

Reviewer #1: No

Reviewer #2: Yes

2. Has the statistical analysis been performed appropriately and rigorously? 

Reviewer #1: N/A

Reviewer #2: Yes

3. Have the authors made all data underlying the findings in their manuscript fully available?

Reviewer #1: Yes

Reviewer #2: No

4. Is the manuscript presented in an intelligible fashion and written in standard English?

Reviewer #1: No

Reviewer #2: Yes

5. Review Comments to the Author

Reviewer #1: 1. This manuscript conducted a review on Community Interventions in Low- and Middle- Income Countries to Inform COVID-19 Control Implementation Decisions based on literature of SARS and influenza. The topic of this paper fall within the scope of the journal. However, since largist review of evidence had demonstrated that distancing and masks cut COVID-19 risk, the scientific value is limited and of little significance for conducting a review on community interventions to Inform COVID-19 control implementation decisions, especially based on literature of SARS and influenza.

2. The authors need to be more cautious to draw a conclusion. The authors mentioned in the Abstract that:

“The evidence confirms the use of face masks, good hand hygiene and social distancing as community intervention that are effective to control the spread of Covid-19 in LMIC.”

However，from the existing evidence (results) of this paper, we can only conclude that the use of masks, good hand hygiene and social distance and other community intervention measures are effective for the control of SARS and influenza. Although covid-19, SARS and influenza are all respiratory infectious diseases, their infectivity is different. It is not appropriate to infer from covid-19 effective community interventions for SARS and influenza.

3. Some statements in this paper are lack of data/reference support. For example, page 8, the authors need to provide data or references to support the following statement “Community-wide face masking has been shown to control the incidence of COVID-19 in Hong Kong Special Administration region compared to other countries”.

4. General scientific writing should be improved throughout the manuscript. For example, page 11, the authors mentioned “Despite the mitigation implementation, there is still a rise in the number of cases per day, as per the daily Kenyan Government briefing on the status of the situation.” The authors should point out the time point this refers to.

Reviewer #2: This is a study conducted as systematic review on community intervention on corona virus 2019, taking a cue on how similar virus spread were controlled .

The study is technically sound and elaborate enough to bring a reduction in the spread of the disease if it's recommendations are implemented.

The study design is just on point, however the following minor comments should be considered:

Fig 7 and 8 , i.e. on innovative hand washing and social distancing cannot be fully traced by the link provided. Only the grid of the small girl can be found by the link. So make the appropriate references on that and provide a full link to all the pictures in the work.

The authors conducted a subgroup analysis based on the type of viruses the previous works that were used for the Meta analysis. However in filling the prisma guidelines in the supplementary section, the authors said such was not applicable to this study. Could the authors explain to potential readers why such is not applicable?

The author will need to tell readers why a statistical test for bias wasn't possible. Also a sensitivity analysis would be necessary to inform the scientific community that, even if a bias test wasn't possible, but the reported pooled effect were not influence by non precise studies in the analysis

The authors reported " LMIC countries" in various sections of the main manuscript. It should just be written as LMIC, since the "C" ,in the acronym stands in for countries.

An excel file of the minimum data used in the analysis should be included in the supplementary section.

It should be accepted for publication with minor correction

6. PLOS authors have the option to publish the peer review history of their article (what does this mean?). If published, this will include your full peer review and any attached files.

Reviewer #1: No

Reviewer #2: **Yes: **JULIUS ABESIG

---

## [Author Response · Author response to Decision Letter 0]

21 Oct 2020

Rebuttal letter: response to reviewers

PONE-D-20-21265

Community Interventions in Low- and Middle- Income Countries to Inform COVID-19 Control Implementation Decisions in Kenya: A Rapid Systematic Review

PLOS ONE

Journal Requirements: 

 Thank you for the guidance, the PLOS ONE’s style requirement have been incorporated accordingly in all sections.

2. We note that the results of your quality assessment using the Cochran and Robins scales are not shown. Please include a table showing how each study scored on each item of these scales. moreover, please conduct a publication bias assessment. The results of quality assessment using the Cochran and Robins scales have now been incorporated into the list of figures. Thanks 

 With support from the communication experts at AFIDEP, we have copy edited the whole manuscript for language usage, spelling and grammar according. The communication experts within AFIDEP with leadership from Elizabeth Kahurani supported with the process. Thank you

We will update your Data Availability statement to reflect the information you provide in your cover letter. Thank you for the guidance. Since the study is a systematic review we have included in the supplementary material a data extraction form that guided the authors in extracting key characteristics of the studies.

In addition, I have included the minimal data set used to reach the conclusion in the manuscript as a separate attachment called ‘supporting information’. Thanks

[This review was resourced through the Heightening Institutional Capacity for Government Use of

Health Research (HIGH-Res) project, which is funded the Alliance for Health Policy and Systems

Research (AHPSR).]

 [The funders had no role in study design, data collection and analysis, decision to publish, or preparation of the manuscript.]

Please include your amended statements within your cover letter; we will change the online submission form on your behalf. Thanks you, we have deleted the funding information in the acknowledgment section.

In addition, I have deleted any related funding related text in the manuscript.

Rather, I have included the amended statements within the cover letter as instructed.

In the online submission form I would like to update the funding statement to read;

The study is funded under Heightening Institutional Capacity for Government Use of Health Research (HIGH-Res) Project. The HIGH-Res project received financial support from the Alliance for Health Policy and Systems Research (the Alliance) at the World Health Organisation, and Wellcome Trust. The funders had no role in study design, data collection and analysis, decision to publish, or preparation of the manuscript.

6. PLOS requires an ORCID iD for the corresponding author in Editorial Manager on papers submitted after December 6th, 2016. Please ensure that you have an ORCID iD and that it is validated in Editorial Manager. To do this, go to ‘Update my Information’ (in the upper left-hand corner of the main menu), and click on the Fetch/Validate link next to the ORCID field. This will take you to the ORCID site and allow you to create a new iD or authenticate a pre-existing iD in Editorial Manager. Please see the following video for instructions on linking an ORCID iD to your Editorial Manager account: https://www.youtube.com/watch?v=_xcclfuvtxQ The ORCID Id for the corresponding author has been created. The ORCID no. is 0000-0002-7198-0558

Thanks this is noted and the supporting information files have been named accordingly.

8. We note you have included a table to which you do not refer in the text of your manuscript. Please ensure that you refer to Table 2 in your text; if accepted, production will need this reference to link the reader to the Table. Thank you, we have ensured that we refer to Table 2 in the text

9. Please ensure that you refer to Figure 8 in your text as, if accepted, production will need this reference to link the reader to the figure. Thank you, we have referred to all the figures into the text. Through realignment we don’t have fissure 8 at the moment.

10. We note that Figure [6 and 7] includes an image of a patient / participant in the study. 

 Thank you, due to lack of consent, we have removed the figure 6&7 and any other textual identifying information or case descriptions for this individual.

Reviewers comments: overall comment to look improve 

1. Is the manuscript technically sound, and do the data support the conclusions?

Reviewer #1: No

Reviewer #2: Yes

 The manuscript is revised accordingly to improve its scientific writing. The conclusions are revised as well to reflect the data presented. Thank you

2. Has the statistical analysis been performed appropriately and rigorously?

Reviewer #1: N/A

Reviewer #2: Yes

 Statistical analysis has been performed appropriately with subgroup analysis indicated where relevant. A minimum data have been provided in the attached document called ‘supporting materials’ to elaborate the number of interventions and control and where subgroup was applied.

3. Have the authors made all data underlying the findings in their manuscript fully available?

Reviewer #1: Yes

Reviewer #2: No

 Minimum data on the study have been included in a separate document attached as ‘supporting information’

4. Is the manuscript presented in an intelligible fashion and written in standard English?

Reviewer #1: No

Reviewer #2: Yes

 The manuscript has been reviewed again and all the grammatical error have been corrected. Thank you

Reviewers' comments: Reviewer #1: 

1. This manuscript conducted a review on Community Interventions in Low- and Middle- Income Countries to Inform COVID-19 Control Implementation Decisions based on literature of SARS and influenza. The topic of this paper fall within the scope of the journal. However, since largist review of evidence had demonstrated that distancing and masks cut COVID-19 risk, the scientific value is limited and of little significance for conducting a review on community interventions to Inform COVID-19 control implementation decisions, especially based on literature of SARS and influenza. The study confirms the use of face masks, good hand hygiene and social distancing as community intervention that are effective to control the spread of respiratory conditions like SARS and Influenza in LMIC. Even though there are a number of emerging studies on Covid-19 our study elaborated with examples from LMIC context on how community interventions have been used and implemented to combat COVID-19. Hence, the importance of the study.

2. The authors need to be more cautious to draw a conclusion. The authors mentioned in the Abstract that:

“The evidence confirms the use of face masks, good hand hygiene and social distancing as community intervention that are effective to control the spread of Covid-19 in LMIC.”

However from the existing evidence (results) of this paper, we can only conclude that the use of masks, good hand hygiene and social distance and other community intervention measures are effective for the control of SARS and influenza. Although covid-19, SARS and influenza are all respiratory infectious diseases, their infectivity is different. It is not appropriate to infer from covid-19 effective community interventions for SARS and influenza. Thanks for the inputs. We have revised the abstract to read “The evidence confirms the use of face masks, good hand hygiene and social distancing as community intervention that are effective to control the spread of SARS and influenza in LMIC.”

3. Some statements in this paper are lack of data/reference support. For example, page 8, the authors need to provide data or references to support the following statement “Community-wide face masking has been shown to control the incidence of COVID-19 in Hong Kong Special Administration region compared to other countries”. The manuscript is revised to ensure proper referencing have been provided and that included the statement in page 8 as mentioned. Thanks

4. General scientific writing should be improved throughout the manuscript. For example, page 11, the authors mentioned “Despite the mitigation implementation, there is still a rise in the number of cases per day, as per the daily Kenyan Government briefing on the status of the situation.” The authors should point out the time point this refers to. Thank you. The manuscript is revised accordingly to improve the scientific writing and referenced accordingly.

On page 11, the sentence is para-praised to read “As of July 2020, the Kenyan Ministry of Health in the daily Kenyan Government briefing reported a rise in the number of cases per day despite the mitigation implementation”. 

Reviewers' comments: Reviewer #2: 

1. The study is technically sound and elaborate enough to bring a reduction in the spread of the disease if it's recommendations are implemented.

The study design is just on point, however the following minor comments should be considered:

Fig 7 and 6 , i.e. on innovative hand washing and social distancing cannot be fully traced by the link provided. Only the grid of the small girl can be found by the link. So make the appropriate references on that and provide a full link to all the pictures in the work. Thank you, due to lack of consent, we have removed the figure 6&7 and any other textual identifying information or case descriptions for this individual.

2. The authors conducted a subgroup analysis based on the type of viruses the previous works that were used for the Meta analysis. However, in filling the Prisma guidelines in the supplementary section, the authors said such was not applicable to this study. Could the authors explain to potential readers why such is not applicable? Apologies that was a typo, we agree subgroup analysis was conducted. The correction with page number is indicated in the prima guideline accordingly.

3. The author will need to tell readers why a statistical test for bias wasn't possible. Also a sensitivity analysis would be necessary to inform the scientific community that, even if a bias test wasn't possible, but the reported pooled effect were not influence by non-precise studies in the analysis We have added a section ‘Sensitivity analysis and assessment of reporting biases’ where we have explained why the test wasn’t possible. Thank you

4. An excel file of the minimum data used in the analysis should be included in the supplementary section. Thank you. We have added the minimum data used in the analysis as separate attachment called ‘supporting information’

---

## [Decision Letter · Decision Letter 1]

3 Nov 2020

Community Interventions in Low- and Middle- Income Countries to Inform COVID-19 Control Implementation Decisions in Kenya: A Rapid Systematic Review

PONE-D-20-21265R1

Dear Dr. Leila H Abdullahi

We’re pleased to inform you that your manuscript has been judged scientifically suitable for publication and will be formally accepted for publication once it meets all outstanding technical requirements.

Kind regards,

Francesco Di Gennaro

Academic Editor

PLOS ONE

Additional Editor Comments (optional):

Dear Authors congratulations

Reviewers' comments:

Reviewer's Responses to Questions

**Comments to the Author**

1. If the authors have adequately addressed your comments raised in a previous round of review and you feel that this manuscript is now acceptable for publication, you may indicate that here to bypass the “Comments to the Author” section, enter your conflict of interest statement in the “Confidential to Editor” section, and submit your "Accept" recommendation.

Reviewer #1: All comments have been addressed

Reviewer #2: All comments have been addressed

2. Is the manuscript technically sound, and do the data support the conclusions?

Reviewer #1: Yes

Reviewer #2: Yes

3. Has the statistical analysis been performed appropriately and rigorously? 

Reviewer #1: Yes

Reviewer #2: Yes

4. Have the authors made all data underlying the findings in their manuscript fully available?

Reviewer #1: Yes

Reviewer #2: Yes

5. Is the manuscript presented in an intelligible fashion and written in standard English?

Reviewer #1: Yes

Reviewer #2: Yes

6. Review Comments to the Author

Reviewer #1: (No Response)

Reviewer #2: This was a revision of an original submission. The authors have addressed all the issues i raised. I have no other comments to the authors. The paper falls in line with the journal scope. it should be accepted for publication .

7. PLOS authors have the option to publish the peer review history of their article (what does this mean?). If published, this will include your full peer review and any attached files.

Reviewer #1: No

Reviewer #2: No

---

## [Editor Report · Acceptance letter]

19 Nov 2020

PONE-D-20-21265R1 

Community Interventions in Low- and Middle-Income Countries to Inform COVID-19 Control Implementation Decisions in Kenya: A Rapid Systematic Review 

Dear Dr. Abdullahi:

I'm pleased to inform you that your manuscript has been deemed suitable for publication in PLOS ONE. Congratulations! Your manuscript is now with our production department. 

Kind regards, 

on behalf of

Dr. Francesco Di Gennaro 

Academic Editor

PLOS ONE